# Insect Biodiversity in a Prealpine Suburban Hilly Area in Italy

**DOI:** 10.3390/insects14090727

**Published:** 2023-08-24

**Authors:** Daniela Lupi, Adriano Zanetti, Paolo Triberti, Sergio Facchini, Fabrizio Rigato, Costanza Jucker, Serena Malabusini, Sara Savoldelli, Paolo Cortesi, Augusto Loni

**Affiliations:** 1Department of Food, Environmental and Nutritional Sciences, University of Milan, Via Celoria 2, 20133 Milano, Italy; costanza.jucker@unimi.it (C.J.); serena.malabusini@unimi.it (S.M.); sara.savoldelli@unimi.it (S.S.); paolo.cortesi@unimi.it (P.C.); 2Museo Civico di Storia Naturale, Lungadige Porta Vittoria, 9, 37129 Verona, Italy; adriano.zanetti50@tiscali.it (A.Z.); caloptilia2@gmail.com (P.T.); 3Sergio Facchini, Via Prati 12, 29121 Piacenza, Italy; sfacchini@virgilio.it; 4Museo Civico di Storia Naturale, Corso Venezia 55, 20121 Milano, Italy; fabrizio.rigato@comune.milano.it; 5Department of Agriculture, Food and Environment, University of Pisa, Via del Borghetto 80, 56126 Pisa, Italy; augusto.loni@unipi.it

**Keywords:** checklist, anthropic areas, woodlands, meadows, monitoring methods

## Abstract

**Simple Summary:**

In the present work, we considered areas located in a park of 4000 ha in northern Italy in a suburban area, with an altitude between 190 and 960 m asl. Our aim was to update the list of insect species and to evaluate the influence of habitat and the locality on the beetle fauna present in the park. We therefore monitored the insects using different collection techniques. A total of 409 different species were collected. Species in Coleoptera were the most abundant, followed by Lepidoptera and Hymenoptera, with other orders in smaller numbers. The analysis showed that the habitat influenced the communities more than the locality. Unfortunately, as there is no detailed historical information on the presence of the species in the area, we cannot hypothesize an effect of the area on the communities. This research, therefore, allowed us to list numerous species for the area, among which some are important, as they were never detected in this area, and constitutes a new starting point for future research on the biodiversity in the prealpine areas.

**Abstract:**

Human activities and habitat fragmentation are known to greatly influence biodiversity. The aim of this study was to update an entomological checklist of a prealpine area in Italy, and also to evaluate the influence of different habitats and the proximity to cities on the entomological fauna. This study included different areas of a local park in Northern Italy, covering about 4000 ha, and situated at altitudes between 190 and 960 m asl. The surveys were carried out between 2010 and 2013 using different monitoring techniques (pitfall traps, car mounted nets, light traps, direct catches on soil and vegetation, visual sampling, gall collection). Furthermore, to assess the effect of habitat and locality on the composition of epigeic beetles, pitfall traps were set and inspected from April to September. All captured specimens were classified to species level. A total of 409 species were recorded, belonging to 7 orders and 78 families. A total of 76.1% were represented by Coleoptera, 13% Lepidoptera, 9.4% Hymenoptera, followed by other orders. In particular, some species with peculiar characteristics, or whose presence in the area had not been previously reported, were detected, such as *Atheta pseudoelongatula*, *Ocypus rhaeticus*, *Tasgius tricinctus*, *Euplagia quadripunctaria*, *Scotopteryx angularia*, *Elachista constitella*, *Parornix bifurca*, *Oegoconia huemeri*, and *Lasius (Lasius) alienus*. It seems possible that the habitat affected the community more than the locality. The woods showed a reduced biodiversity, and a simplified community structure. The comparison of the same habitats in different localities did not show significant differences.

## 1. Introduction

In Europe, and also in Italy, human activities such as grazing or controlled deforestation for firewood have modified the environment in recent centuries, allowing the coexistence of woods and meadows in the hills, but also allowing the establishment of stable populations of arthropods over the centuries [1,2]. However, since the middle of the 20^th^ century, these activities have been abandoned, with further modifications in the areas affected by the change. On the one hand, meadows, without the selective action of grazing, have been reduced and closed to give way to forests [3,4]. On the other hand, unmanaged forests have changed their structure, allowing the colonization of more invasive plant species such as *Robinia pseudoacacia* L. (Magnoliopsida, Fabaceae) and *Ailanthus altissima* (Mill.) Swingle (Magnoliopsida, Simaroubaceae) have rapidly replaced the native plants that previously characterized the woods [1,5,6,7]. As a direct consequence of such sudden and severe habitat modification, arthropod populations may be altered [8,9], with many species becoming rare, endangered, or threatened [10,11,12], and cosmopolitan or exotic invasive species settling [13,14].

Therefore, there is a worldwide concern regarding the loss of biodiversity [15], and there is also an increasing need to evaluate the role that different habitats play in ensuring a certain balance and maintaining existing biodiversity [16,17,18]. As human activities have put many species at risk, the International Union for Conservation of Nature has emphasized the need to protect existing environments (natural, seminatural, and even “man-made” areas) from further change and destruction [18,19,20]. For this reason, several studies have focused on the effects of urbanization on biodiversity [21,22]. Although much research has shown the role of urbanization in biodiversity decline [23,24], there is also evidence that fragments of wooded urban habitats can even host several endemic species, although the abundance and richness of the species they contain decreases as the level of human disturbance increases [20,21]. However, the situation is also complicated by the fact that for many species, there is no information on their status, and historical data are also lacking. A review [12] has demonstrated that the countries localized in the southern part of Europe are the richest in number of species, but are also the ones in which there is the least information about the current presence and abundance of species. 

Northern Italy, at the center of Italian industry, is characterized by intensive agriculture and few seminatural areas, which can be considered the primary reserves of biodiversity. However, in zones closely linked to densely populated and industrialized cities, natural areas or periurban parks are more vulnerable, and anthropic perturbations puts their biodiversity at risk. The aim of the present research is to evaluate the richness of insect species of a hilly wooded area included in a park in the Po Plain surrounding the city of Brescia and its hinterland (Northern Italy), since it can be considered representative of similar areas in northern Italy. This park, which extends over several municipalities, is considered a suburban natural reserve. According to Italian regulations (Regional Rule l.r 1986/83 art.34), it is registered as a Park of Supramunicipal Interest (P.L.I.S.) (a form of protection established by the Lombardy Region that allows municipalities to identify and safeguard areas of environmental and naturalistic importance). As the information on the biodiversity of insects in this area is limited to studies from caves, and to some scarce information inserted in the Italian checklist [25], and that there is no historical dataset, the first objective was to create an updated record of insect fauna in the area. Furthermore, as the area represents a path of connection and integration between urban and natural areas, and includes several different seminatural habitats, the aim was to evaluate the importance that the different environments have in supporting the biodiversity actually present in them, also given their proximity to crowded, industrialized cities. For these reasons, two areas characterized by the presence of different habitat were selected, where different monitoring techniques were applied. Considering the impossibility of collecting all the species of an area, and even of a single taxonomic group, even with several sampling methods (both trapping and visual observations) [26], the present work represents an important step in obtaining a representative idea of the current species richness of an area of the Alpine arc. In addition, the assessment of the possible presence of endemics, rare species, and even invasive species, is of paramount importance. Finally, the area can also be representative of similar areas that have been subjected to numerous anthropic actions over the centuries. 

## 2. Materials and Methods

### 2.1. The Study Area

The study was carried out in two different areas (Mount Maddalena and the area of Mount Picastello) of a park called “Parco delle Colline”, located in Brescia Province (Northern Italy). These areas are located at a distance of 10 km from each other. Mount Maddalena was chosen, as it is located in the upper part of Brescia, and hypothetically is subjected to a greater anthropic pressure than Mount Picastello, close to the small municipality of Collebeato, and hereafter referred to as Maddalena and Collebeato. The park covers an area of about 4000 ha, is 190 to 960 m above sea level, and includes territory of the municipality of Brescia, and four other municipalities very close to the city of Brescia. The park’s landscape is characterized by hilly environments, bordering an urban area of about 200,000 inhabitants. About 70% of the park territory is covered by forests, which in the past, were commonly exploited for the production of wood for firewood; the remaining 30% is covered by meadows and clearings. The geographical relief of the park was formed between the end of the Triassic period, the beginning of the Jurassic period (about 200 million years ago), and the Miocene period (about 20 million years ago). The slopes of the hills are furrowed by gullies, by long cliffs, and bare rocky ridges, which correspond to the more resistant limestone rocks. According to Köppen–Geiger classification, the climate in this area is Cfa ((c) temperate, (f) without dry season, (a) with hot summer). However, this classification is very generic, and the area is characterized by different microclimates due to the variability of exposure, altitude, and humidity. For example, on the more exposed and southern slopes, characterized by fewer days of frost and fog than in the areas closest to the plain, we can find typical sub-Mediterranean vegetation, while, on the north-facing slopes and in the areas at higher altitudes, there is a typically mountainous climate. In this context we can find both arboreal and shrubby species of the thermophilic forest such as the downy oak (*Quercus pubescens* Willd), the black hornbeam (*Ostrya carpinifolia* Scop.), and the manna ash (*Fraxinus ornus* L.), as well as the mesophilic forest characterized by species such as the chestnut (*Castanea sativa* Mill.), oak (*Quercus ilex* L. and *Quercus robur* L.), and the hornbeam (*Carpinus* spp.). The area is also characterized by the presence of the black locust (*Robinia pseudoacacia* L.) [27]. 

The two selected areas (Maddalena and Collebeato) are located north-east and north-west of the city of Brescia, respectively. On Maddalena, characterized by a higher altitude (up to 800 m), five localities were selected, and on Collebeato, where the maximum altitude is about 400 m, two were selected. Meadows, woods, and areas with mixed vegetation were selected, if present (Table 1).

### 2.2. Checklist Update 

As the information about the entomological fauna in the considered area is limited to a list of 34 species reported from Mount Maddalena [25], to update the checklist of the insect taxa, a census was carried out from 2010 to 2013 in the area (specifically Maddalena, and Collebeato). The following monitoring techniques were used: pitfall traps, car mounted net, light traps, direct capture on soil and vegetation, visual sampling, and gall collection. 

Coleoptera were monitored with pitfall traps, car mounted nets, cross-vane traps, and direct captures on soil and vegetation; Lepidoptera with direct capture using a light-trap and direct captures on vegetation; Diptera with direct capture on vegetation and soil, and Hymenoptera with pitfall traps, direct capture, and gall collection. Details of the different methods are given below. Regarding forests, different habitats were explored: mature (mixed deciduous trees and chestnut woods) and young (new chestnut planting) forests, and a newly deforested area. 

Insects collected using different monitoring techniques were prepared and classified at the species level listed in a dataset to obtain a checklist updated with information on the locality (site, altitude, and GPS coordinates), the time of monitoring, and the monitoring method adopted. Due to the differences in the monitoring methods, information on the number of specimens collected was not added to the checklist, and each species was inserted only once per monitoring and per locality. In the classification of the species of particular interest, the chorology of the species was obtained from Authors’ personal data, or from the checklist of the species of Italian fauna [28,29,30]. The nomenclature was also according to the latter works. When necessary, species were compared with those in the Authors’ private collections, or in university or museum collections. The identification of Macrolepidoptera was mainly carried out with [31,32].

#### 2.2.1. Pitfall Traps 

Pitfall traps (7 cm in diameter, 10 cm in depth, covered with a stone and filled with vinegar as a lure to attract beetles, and salt (10%) to preserve the contents from fermentation) were positioned at ground level to check the species of ground-dwelling insects. Ten pitfall traps were positioned in Collebeato, and 25 in Maddalena. Traps were replaced fortnightly, from April to September in 2010, 2011, and 2012. The distance between traps in the same locality ranged from 50 to 100 m. All of the collected specimens were classified at the species level. Carabidae and Staphylinidae were counted and used for community structure analysis (see Section 2.3).

#### 2.2.2. Car-Mounted Net

To capture flying Staphylinidae, a net was mounted over the roof of a car according to [33]. The air in which the rovebeetles fly is forced up and over the hood of the car, allowing the capture of the specimens in the net [34]. The trap consisted of an elongated pyramid-shaped net, ending in a detachable bag connected with Velcro. The net was fixed, consisting of a lightweight metal structure (with an entrance opening of about 110 cm × 64 cm), supported by a crossbar attached to the roof of the car. Considering the car used, the net was mounted at a height between 1.60 and 2.20 cm.

The monitoring was made on a single date chosen at the time when most species were thought to be present (15 July 2010), at sunset at low speed (20–30 km/h), covering the traffic road (of about 6 km) that goes from the base to the top of Maddalena four times. 

#### 2.2.3. Cross-Vane Traps 

In 2011, a total of four cross-vane traps (Maddalena *n* = 2; and Collebeato *n* = 2) were positioned from June to September. They were placed at a height of 2 m on plants in two mature deciduous forests, located, respectively, at altitudes of 400 m (Maddalena) and 330 m (Collebeato). The traps were checked fortnightly, and insects were removed and brought to the laboratory for classification. 

#### 2.2.4. Light-Trap Captures

Two different light traps were used for the nocturnal Lepidoptera: a round tower of synthetic fabric trap (2 m height) illuminated from the inside by a 15 W actinic light lamp powered by a 12 V battery, and a white cotton sheet with similar light sources. Observations were made on a single day corresponding with the period when most trapping was expected to occur (14 July 2010). The traps were set up and observed from 8 p.m. to midnight. They were located at an altitude of 750 m in a meadow with a large opening to the surrounding area. 

#### 2.2.5. Visual Sampling and Direct Captures

In order to improve the checklist and to avoid an excessive use of destructive methods, visual detection of insects belonging to different orders, and direct classification in the field, were adopted whenever possible. A total of 19 visits for visual sampling, and 24 for direct captures, were made from March 2010 to July 2013 in Maddalena and Collebeato. The sampling in question had no inferential value, but only the objective of updating the list of species present. In this way, it was not necessary to standardize the timing, the length of the path, and the randomization of sampling in the monitoring. For this reason, the area was monitored by walking along the path over the hill and stopping in front of plants randomly selected during the walk, or when insects were observed. Image-based solutions using cameras (Canon EOS 5D Mark II equipped with Canon EF 100 mm f/2.8L IS USM Macro Lens, Canon, Tokyo, Japan) were adopted whenever a specimen was detected on vegetation and soil. Images were then checked in laboratory for classification, and only images showing specific morphological and color patterns were used to assign a species name to include in the checklist. 

#### 2.2.6. Gall Collection

Data from a survey on chestnuts in Maddalena in 2013 [35], related to monitoring the presence of the invasive chestnut gall wasp *Driocosmus kuriphilus* Yasumatsu (Hymenoptera, Cynipidae) and its natural enemies, were integrated with samples of other gall insects in 2010, 2011, and 2013. Cynipidae classification was based on the shape of the gall and the plant species from which it was collected, according to [36]. Galls were removed and stored separately for each plant and sample, and positioned in aerated rearing boxes under outdoor conditions until the emergence of the adult gall wasps, and possible parasitoids.

### 2.3. Analysis of the Community Structure of Carabidae and Staphylinidae Habitats

Carabidae and Staphylinidae were selected to study the effect of the habitat and localities on the community structure of the epigeic beetles from 2010 to 2012 in Mount Maddalena and in Collebeato. Five pitfall traps per locality and habitat were positioned in Mount Maddalena (*n* = 25) and in Collebeato (*n* = 10) and replaced fortnightly, from April to September in 2010, 2011, and 2012. Distance between traps in the same locality ranged from 50 to 100 m. 

Areas were classified on the basis of the dominant species and woods (W), meadows (M), and mixed vegetation (MW) were present and selected in Mount Maddalena, while meadows (M) and mixed vegetation (MW) were in Collebeato (Table 1). Monitoring sites were selected from the base of the hill to the top, at altitudes between 350 and 750 m in Mount Maddalena, and around 330 m in Collebeato. 

The experimental conditions adopted in this part of the study resulted in a mixed, spatially nested design, with four factors: the fixed factor “district” with two levels, Maddalena (M) and Collebeato (C). These two levels are characterized by a different pressure of anthropic activity. Nested in this factor remains the fixed factor “locality” with 7 levels, five within the level M (M1–M5), and two within the level C (Cc and Pn). The levels of this factor were analyzed on the basis of the vegetational structure dominant species of wood (W), meadow (M), and mixed vegetation (MV) (Table 1). In each locality, we chose the random factor “site” with two levels, 1 and 2. At each site we installed five pitfall traps, which were placed approximately at a distance of about 15–20 m apart and replaced fortnightly, from April to September.

The data of the multiyear captures have been pooled together (no year factor) to obtain a more complete and reliable dataset of the Carabidae and Staphylinidae populations of the studied areas, avoiding the obvious factor of variability due to the time and the natural fluctuations of species populations [37].

The data have been organized into a raw data matrix. We applied a log(X + 1) transformation to the raw data to avoid the right-skewed distribution of the species, and calculated the Bray–Curtis similarity coefficient, the most used multivariate index in biodiversity community structure studies [38]; we obtained a similarity matrix that reports the percentage similarity values among all pairs of samples.

On this matrix, we applied the ordination principal coordinates analysis (PCO). It produces a perpendicular projection of the sample points, directly on the axes, maximizing the variance of the projected sample points. 

To explore the potential relationships between the sets of our variables (species) and the ordination axes, we superimposed the vectors of the most represented species on the axes, expressing the correlations of the vectors themselves with the PCO axes. 

We tested the null hypothesis (that no differences exist among the composition of the sample groups of the Carabid population) by the nonparametric permutational, multivariate analysis of variance (Permanova), using the mixed, nested, spatially hierarchical design, described above. We chose a significance p level of 0.001. Such an experimental design was unbalanced, due to the different number of replicates between the two districts, so we adopted a “sequential” type I sum of square, more appropriate for nested, hierarchical models [39].

The homogeneity of the data dispersion was assessed by the permutational analysis of data dispersion (Permdisp), applied on all the three factors of the design. 

We also studied the level of biodiversity among the communities of different localities by calculating the diversity index of Shannon–Wiener (H’) and the Pielou’s evenness index for all the localities, and by grouping the data with the same main habitat factor (wood, meadow, and mixed vegetation). 

Simper analysis was then used to identify the species that contributed to more than 90% of the average similarity in the different habitat groups.

All of these analyses were performed by using the software package Primer v6 (Primer-E Ltd., Plymouth, UK), user manual/tutorial and Permanova+ for Primer were used for all analysis [38,39,40].

## 3. Results

### 3.1. Checklist Update 

The data of all species detected during the monitoring period are reported in a dataset [41].

Considering that different taxa were sampled with different intensity in relation to the methods used, the values presented reflect the protocol. In detail, a total of 409 species belonging to 7 orders and 78 families were classified during the monitoring period (Table 2). Among the species identified, the order Coleoptera was the most represented (76.1%), followed by Lepidoptera (13%), Hymenoptera (9.4%), and Hemiptera (1.02%). Diptera represented 0.03% of the total number of species detected, and Orthoptera and Neuroptera 0.08%. Considering Coleoptera, Lepidoptera, and Hymenoptera, the three orders most caught, a total of 26, 30, and 9 families, respectively, were identified. 

Regarding the trapping methods, pitfall traps covered 77.18% of the captures, followed by light traps (10.12%), visual sampling (5.37%), and the others (7.33%).

#### 3.1.1. Coleoptera

Coleoptera was the most numerous taxon, covering 49.38% of the species detected, with 202 species. Obviously, this high number is biased by the sampling methodology adopted in the specific part of the study, focused on the epigeic beetle community structure.

Carabidae and Staphylinidae were the most represented families, accounting for 67.33% of the species of Coleoptera found. Curculionidae and Cerambycidae, both with 13 species, accounted for 12.88% of the Coleoptera, while the remaining 19.79% was represented by 22 families, some with only 1 species per family. 

Carabid beetles included 41 species, and the genus *Abax* was the most abundant, with three species covering 32% of the Carabidae identified, (*Abax (Abax) parallelepipedus lombardus* Fiori, *Abax (Abax) baenningeri* Schauberger, and *Abax (Abax) fiorii* Jacobson).

In the family of Staphylinidae, a total of 95 species were collected, mostly with pitfall traps. In addition, the use of a car net allowed the capture of different species which are difficult to collect with pitfall traps. The genus *Atheta* covered 41% of those detected, and the species detected the most frequently were: *Atheta triangulum* (Kraatz), *Atheta crassicornis* (Fabricius), *Atheta oblita* (Erichson), and *Atheta* gr. *trinotata*. *Atheta pseudoelongatula* Bernhauer, which was collected with the car net on 15/07/2010, is an exotic species already detected by [42] in a nearby region.

Species in other families were detected mostly in few localities and on few occasions, except for the Scarabeidae *Protaetia (Netocia) morio* (Fabricius) and the Silphidae *Silpha carinata* (Herbst), which were found in different sites and years in Maddalena and Collebeato. 

#### 3.1.2. Lepidoptera 

Lepidoptera were collected with light traps or observed by visual sampling on vegetation.

A total of 150 species were classified, belonging to 30 different families covering 36.67% of species detected. A total of 93% of them were collected on Mount Maddalena, from the trap light monitoring. Major captures belonged to families Noctuidae (16.00%) with 24 species, Geometridae (14.67%) with 22 species, and Tortricidae (13.33%) with 20 species. 

#### 3.1.3. Hymenoptera

Hymenoptera covered 10.27% of the species detected (*n* = 42), with 9 families. The family Formicidae, mainly collected with pitfall traps, represented 54.76% of the species of Hymenoptera classified, with 23 species, followed by Apidae and Cynipidae that together represented 23.81%, with 5 species each. The species of Cynipidae emerged from galls collected during the survey were associated with oak, chestnut, and acer trees, and wild rose plants. In the laboratory, Hymenoptera parasitoids also emerged from the galls and, in detail, species belonged to the family Torymidae (*Megastigmus dorsalis* (Fabricius), *Torimus flavipes* (Walker), *T. formosus* (Walker), and *T. sinensis* Kamijo) and Eupelmidae (*Eupelmus urozonus* Dalman). Remarkable is the detection of *Lasius alienus* (Foester) on Mount Maddalena, which is a rare species in Italy. 

#### 3.1.4. Hemiptera, Diptera, Neuroptera, Orthoptera

The species in the orders Hemiptera, Diptera, Neuroptera, and Orthoptera were grouped together, as they represented 3.67% of all the remaining species. They were observed by visual sampling and direct capture at the adult stage. 

A total of nine different species of Hemiptera were detected, as well as four species of Diptera (*Bibio marci* (L.); *Bombylius (Bombylius) major* L.; *Scathophaga* sp.; *Volucella pellucens* (L.)), one species of Neuroptera (*Libelloides coccajus* (Denis et Schiffermuller)), and one of Orthoptera (*Barbitistes serricauda* (Fabricius)).

### 3.2. Analysis of the Community Structure of Carabidae and Staphylinidae 

The results of the ordination analysis are shown in Figure 1. The first two axes of PCO graphical representation explain 30.3% of total samples variation, not giving a strongly reliable interpretation of the data. Nevertheless, it was quite clear that the samples of the locality (M3) and (M4) are more concentrated in the high, right corner of the graphic, and their projections on the two axes are well separated from the samples of the other localities (Figure 1A). The ordination, reproduced by symbolizing the samples on the basis of the factor habitat, shows that the samples which are more concentrated are in the level (W) of the factor “Habitat”, while all the other samples of levels (M) and (MV) were more dispersed, and partially separated on the second axes (Figure 1B).

Such distribution of samples suggested a similarity of samples of the locality (M3) and (M4), which represent the same habitat level (W). Based on this observation, we ran the PERMANOVA analysis by considering the level (W) of the factor “Habitat” in contrast with pairs of the other two levels (MV, M). The PERMANOVA analysis showed significant statistical differences for the factors district and locality, but not for the factor habitat, and differences emerged in the “Contrast”, confirming the separation of the groups highlighted by the ordination (Table 3). 

The PERMDISP (permutation of dispersion) analysis was also significant for the factors locality and habitat, so the factor effects could be masked by the dispersion of data (group factor: district, number of permutations: 9999, F: 9,6374, df1: 1, df2: 425, P (perm): 0.014; group factor: locality, F: 5.4566, df1: 6, df2: 420, P (perm): 0.002; group factor: habitat, F: 8.2162, df1: 2, df2: 424, P (perm): 0.003). Our interest has focused on the factor habitat “H”, which responds the best to ecological elements for the carabid community, so we performed a pairwise test on such a term. Such analysis confirmed what was described by ordination and by PERMANOVA analysis. The level “W” is different from both “MV” and “M” levels (Table 4).

The only two species that showed a good correlation rate with the graphical distribution of the samples in the ordination were *Atheta trinotata* and *Abax parallelepipedus* (δ = 6.5). A vector of *A. trinotata* develops parallel to the first axis, correlating with the clustering of W habitat samples, while a *A. parallelepipedus* vector develops to the cluster of MV samples.

Table 5 reports the diversity indices calculated on the capture data summed up on the basis of the factors locality and habitat. Localities M3 and M4 had the lowest values of H’, as well as the Pielou evenness (J’). The same results were obtained by calculating an index on the data of the captures summed on the basis of the habitat factor, where the factor “W” showed the lowest values of the H’ and J’ indices. 

The SIMPER analysis was performed on the habitat groups. The average Bray–Curtis similarity between all pairs of samples in the meadow group is 10.67. *Atheta* gr. *trinotata* and *Silpha carinata* together accounted for almost 60% of the average similarity. *A. trinotata* contributed 32.28% with the highest similarity/standard deviation ratio, meaning that this species was well represented and distributed across all samples in the group. *S. carinata* had a lower percentage contribution to the average similarity, as well as a lower similarity/standard deviation ratio, implying a less homogeneous distribution among the samples (Table 6). These two species typified the groups of samples in the M group. The wood group had an average similarity of 17.75, and was typified by the species *A. trinotata*, which alone contributed more than about 60% of the average similarity. This species was also well distributed among the samples (Sim/SD = 0.61). The average similarity of the mixed vegetation group was 16.91, and was typified by the species *Abax parallelepipedus lombardus*, which alone contributed 60.41 of the average similarity, with a high similarity/standard deviation ratio (0.73) (Table 4).

## 4. Discussion 

This research project has allowed us to update information about the communities of insects that are present in a hilly area in the north of Italy close to cities, and probably affected by their proximity. A total of 409 species was detected in the area, with information about the areas where they were detected. In particular, some species with peculiar characteristics, or whose presence in the wider prealpine zone has not previously been reported, were detected, including some Coleoptera and some Lepidoptera. 

All Carabid beetles included 41 species that were very common for the region, and many of them are more typical of the agricultural landscape than of woody areas [43]. Among Staphylinidae, three noteworthy species are included. The first is *Atheta pseudoelongatula* Bernhauer, native to Japan, and detected for the first time in Italy in 2010 in Verona province [42]; in the present study it was collected only with the car net on Monte Maddalena. The other two species are remarkable for their unusual geographical location. *Ocypus rhaeticus* (Eppelsheim) is distributed mostly on the southern slopes of the Alps, although a few records of its presence on the northern slopes exist (North Tyrol and Engadine). It is a typical inhabitant of woodland ground litter, from 300 to 2000 m asl, with an optimum population in the *Fagus* belt. The record shows that *O. rhaeticus* in Monte Maddalena has reached its geographical (southern) and altitudinal (lower) limits. *Tasgius tricinctus* (Aragona) detection confirms its particular geographical distribution, with its main area in the Apennines, and another separate zone in Lombard Prealps, identified as the rare “trans-padan” distributional pattern [44,45].

Most of the Lepidoptera found in the present study are common in Italy, but some of them are noteworthy. Among them, the Erebidae *Euplagia quadripunctaria* (Poda), a species reported for all Italian regions except Sardinia, and known for inhabiting the banks of water courses, marshes, and the edges of riparian woods. The species feeds on *Corylus avellana* L., *Lamium* spp., *Urtica* spp., *Rubus* spp., *Cytisus* spp., and *Eupatorium cannabinum* L., and is inserted in the Annexes II and IV of EU Directive 92/43/EEC within animal species of community interest whose conservation requires the designation of special areas of conservation. Other noteworthy species include the Geometridae *Scotopteryx angularia* (de Villers), quite rare and localized in Italy, associated with dry meadows on hills and mountains (1000 m asl); the Elachistidae *Elachista constitella* Frey, already reported in Italy in Friuli, Venezia Giulia, and Veneto regions in thermophilic woods of downy oak and black hornbeam, but of which the biology is still unknown; the Gracillariidae *Parornix bifurca* Triberti, of which the biology is still unknown, and which is distributed in Basilicata, Abruzzo, Lazio, Toscana, Lombardia, and Piemonte regions; and the Autostichidae *Oegoconia huemeri* Sutter, a species endemic in Malta [46], and for which the larvae are still unknown, and it has been detected up to 1400 m asl. 

Among Hymenoptera, many species detected are common for the Italian Fauna and with a wide distribution in the area. Exceptions were the Cynipidae *Dryocosmus kuriphilus* Yasumatsu and the Torymidae *Torymus sinensis* Kamijo, which are both exotic. *D. kuriphilus* is an invasive gall wasp that has compromised chestnut production in Italy in recent years, while *T. sinensis* is a parasitoid that was successfully used in biological control programs for its control [39]. If we exclude *D. kuriphilus*, the other gall wasps detected are not considered pests. Among Formicidae, it is worth noting the presence of *Lasius (Lasius) alienus* (Foerster), which is a rare species for the area considered, and which was detected on just one occasion in a pitfall trap in one location. Only a few Apoidea (five species) were detected and listed. We are aware that this low number of species collected represents an underestimation of the actual presence of bees. It can probably be ascribed to the monitoring method adopted, and the low presence of flowering species in the areas during the monitoring periods.

In less represented orders, all of the species were common for Italy and Lombardy. In Hemiptera, apart from one specimen in the genus *Rhinocoris* (Family: Reduvidae), which is a well-known predator, all other findings belong to species which are associated with herbaceous or ornamental plants, but are not of particular interest. It is interesting to note the presence of *Libelloides coccajus* (Denis and Schiffermuller), detected only in one occasion in a meadow in Maddalena. It is an euriecious species easy to notice for its dimensions, color, and shape, and is typical of meadows poorly used for grazing. 

The analysis of the epigeic beetle community structure showed a quite simplified ecological community. However, the absence of historical data is noteworthy, as just 34 species were included in the Italian checklist in [25], which poses difficulties in evaluating the change in the communities due to anthropic changes, but our data now provide a basis for further studies in the future. The composition of the beetle community was more influenced by the habitat than by the different localities. As a consequence, as it was assumed that the two places had different anthropic influences due to the urbanization proximity, the comparison of the two localities did not result in significant differences. The differences in the habitats were expected, as they are characterized by different ecotypes, and consequently can host different species. However, the influence of urbanization can be considered widespread in the area, as captures in all localities were characterized by a few dominant species. Other authors have reported that one of the main consequences of anthropic areas is that habitats support less diverse insect communities, confirming the species–area relationship [47]). Furthermore, negative relationships between species richness and patch isolation have also been documented for several insect groups [47,48], due to reduced immigration in the absence of pristine natural habitats in nearby areas. In the present research, the localities characterized by the presence of mixed vegetation and woods showed a reduced biodiversity, and a more simplified community structure dominated by the staphylinid *Atheta* gr. *trinotata* in the wood habitat, and the carabid *Abax parallelepipedus* in mixed vegetation. They really were the most widespread species in both types of habitat, representing generalist, polyphagous species, the most effective in colonizing simplified ecosystems. *A. parallelepipedus* is a nocturnal and polyphagous species, and is often a dominant species in deciduous forest communities [49,50]. *A. trinotata* is a generic predator, often synanthropic, and occurs in any habitat rich in manure and decomposing substrates [51]. The ecologically simplified community of the habitats W and MV, with respect to M, is supported by the contribution of the species *A. trinotata* and *A. parallelepipedus* to the value of the similarity average percentage of the samples of the two habitats. Each species alone contributed to about 60% of this parameter, showing a more unbalanced species abundance distribution, in comparison to the habitat M, where the same percentage value was produced by the contribution of two species. Similarly, 90% of the total average similarity in the W and MV habitats was reached with six species, indicating a more simplified community in comparison to M, which reached 90% with eight species.

## 5. Conclusions 

This work is a pioneering study for similar areas in the prealpine region of Italy. Even if the results are unbalanced towards Coleoptera (due to the methods used), it provides a list of many species living in the area and their dominance/rarity. Many species were dominant, indicating a poorly balanced habitat and low biodiversity, but the detection of species that are rare or typical of few environments in Italy indicates that even though the area has been subjected to anthropic pressure and strong modification for centuries, there are niches that must be preserved from further perturbations to allow the conservation of these species.

## Figures and Tables

**Figure 1 insects-14-00727-f001:**
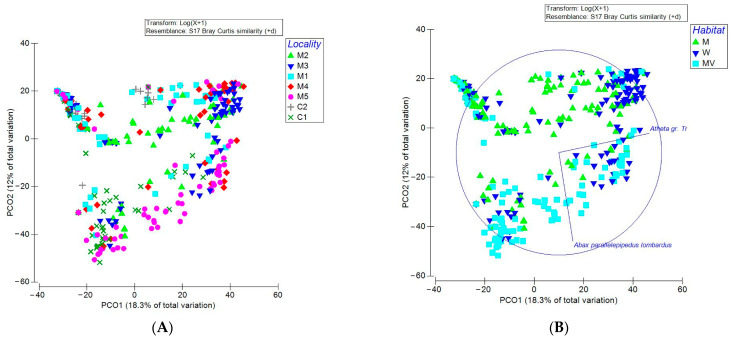
PCO ordination on data of trap samples symbolized according to the locality of sampling (**A**), or according to the habitat represented (**B**).

**Table 1 insects-14-00727-t001:** Vegetal structure in the different localities and habitat classification based on dominant species (listed in order of abundance according to a Phytosociological census).

Locality		Altitude	Habitat Classification	Dominant Species
Maddalena (M)	M1	750–800	Meadow	*Bromus erectus* Hudson
				*Carex humilis* Leyser
				*Dactylis glomerata* L.
				*Rubus ulmifolius* Schott.
	M2	700	Meadow	*Bromus erectus* Hudson
				*Rubus ulmifolius* Schott.
	M3	700	Wood	*Castanea sativa* Mill.
	M4	450	Wood	*Castanea sativa* Mill.
				*Carpinus betulus* L.
				*Quercus petraea* (Matt.)
				*Rubus ulmifolius* Schott.
	M5	450	Mixed Vegetation	*Rosa canina* L.
				*Fraxinus ornus* L.
				*Betula pendula* L.
				*Populus nigra* L.
Collebeato (C)	C1	335	Mixed Vegetation	*Quercus pubescens* Willd.
				*Fraxinus ornus* L.
				*Rosa canina* L.
				*Rubus ulmifolius* Schott.
	C2	320	Meadow	*Bromus sterilis* L.
				*Phleum phleoides* (L.) Karsten
				*Bromus erectus* Hudson
				*Carex caryophyllea* La Tourr.
				*Rubus ulmifolius* Schott.

**Table 2 insects-14-00727-t002:** Number of the species detected according to the order, families, and number of species.

Order	Families	No. of Species	Number of Times That the Species Was Detected
Orthoptera	1	1	1
Neuroptera	1	1	1
Hemiptera	7	9	13
Lepidoptera	30	150	165
Coleoptera	26	202	970
Diptera	4	4	4
Hymenoptera	9	42	120
Total	78	409	1274

**Table 3 insects-14-00727-t003:** Output of the PERMANOVA analysis carried out to test for differences. df = degrees of freedom; SS = sum of squares; MS = mean square; pseudo-F = F statistic; P (perm) = probability levels obtained by permutations. *** *p* < 0.001.

SOURCE	DF	SS	MS	PSEUDO-F	P (PERM)	PERMS
**HA**	2	1.0122E5	50611	1.3698	0.0354	9916
**CONTRAST (W) V (MV, M)**	1	38392	38392	11.244	0.0001 ***	9909
**DI**	1	50422	50422	16.789	0.0001 ***	9930
**LO (DI)**	5	1.1536E5	23071	7.522	0.0001 ***	9943
**SI (LO (DI))**	7	20864	2980.5	0.92968	0.6649	9815
**HA** **×** **SI (LO (DI))**	1	4935.5	4935.5	1.5395	0.096	9936
**RES**	410	1.3145E6	3206			
**TOTAL**	426	1.6073E6				

**Table 4 insects-14-00727-t004:** Results of the pairwise test applied to different habitats. (MV = mixed vegetation; W = wood; M = meadow). T = square root of pseudp-F, *** *p* ≤ 0.001.

GROUPS	T	P (PERM)	UNIQUE PERMS
**MV, W**	3.2345	0.001 ***	9999
**MV, M**	1.266	0.04	9966
**W, M**	3.816	0.001 ***	9998

**Table 5 insects-14-00727-t005:** Diversity indices calculated on data summed for localities and habitats. (S = no. of species; N = no. of specimens; J’ = Pielou evenness index.; H = Shannon–Wiener index).

		S	N	J’	H (Log e)
Locality	M1	30	388	0.61	2.10
	M2	45	1309	0.44	1.67
	M3	58	3754	0.24	0.90
	M4	35	614	0.40	1.44
	M5	54	1060	0.54	2.20
	C1	47	437	0.71	2.73
	C2	22	110	0.82	2.53
Habitat	Mixed Vegetation	70	1807	0.51	2.16
	Wood	68	4368	0.26	1.10
	Meadow	75	1497	0.60	2.58

**Table 6 insects-14-00727-t006:** Simper analysis: species that contributed to more than 90% of average similarity in the different habitat groups (the average Bray–Curtis similarity between all pairs of samples in the group meadow, wood, and mixed vegetation).

Habitat	Species	Average Abundance	Average Similarity	Similarity /SD	Contribution %	Cumulative %
Meadow	*Atheta* gr. *trinotata*	0.66	3.44	0.37	32.28	32.28
	*Silpha carinata*	0.55	2.88	0.26	26.99	59.28
	*Atheta oblita*	0.19	0.92	0.17	8.58	67.86
	*Hister quadrimaculatus*	0.13	0.73	0.11	6.80	74.66
	*Carabus (Tomocarabus) convexus convexus*	0.18	0.58	0.14	5.45	80.11
	*Abax parallelepipedus lombardus*	0.15	0.47	0.15	4.45	84.56
	*Liparus dirus*	0.08	0.38	0.08	3.56	88.11
	*Omalium rivulare*	0.26	0.30	0.12	2.81	90.92
Woods	*Atheta* gr. *trinotata*	1.70	10.59	0.60	59.70	59.70
	*Silpha carinata*	0.58	2.93	0.30	16.51	76.22
	*Abax parallelepipedus lombardus*	0.26	1.04	0.22	5.86	82.07
	*Omalium rivulare*	0.43	0.75	0.22	4.25	86.32
	*Echinodera capiomonti*	0.08	0.39	0.08	2.21	88.53
	*Kyklioacalles aubei*	0.07	0.37	0.08	2.08	90.61
Mixed Vegetation	*Abax parallelepipedus lombardus*	1.07	10.21	0.73	60.41	60.41
	*Atheta* gr. *trinotata*	0.67	3.37	0.38	19.94	80.35
	*Abax baenningeri*	0.24	0.63	0.20	3.72	84.07
	*Drusilla canaliculata*	0.23	0.51	0.16	3.03	87.10
	*Omalium rivulare*	0.25	0.35	0.14	2.07	89.18
	*Protaetia morio*	0.04	0.22	0.05	1.31	90.49

## Data Availability

The list of the species detected is available in the dataset Lupi, D.; Zanetti, A.; Triberti, P.; Facchini, S.; Rigato, F.; Jucker, C.; Savoldelli, S.; Malabusini, S.; Cortesi, P.; Loni, A., “check list Parco colline Lupi et al-1.tab”, Check list Parco colline Lupi et al., https://doi.org/10.13130/RD_UNIMI/S8MW9P/SUWN7Z (accessed on 12 July 2023) UNIMI Dataverse, V3, UNF:6:Iku+CT1F889OD8k6rMw9Xg== [fileUNF].

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
