# Peer review of "Insect Biodiversity in a Prealpine Suburban Hilly Area in Italy"

_insects, 2023, doi:10.3390/insects14090727_

Round 1

Reviewer 1 Report

The language needs to be clear. The authors should tell the readers what was done, and avoid the global scale problem list, for example the opening sentences. This is a good study with lots of excellent work, but the authors do not tell the readers what they found! For example in Results, What species were caught? Other than a few species that are listed in Table 4, the reader has no idea what was caught. The species listed in table 4, what orders are they? The analyses may be appropriate, or not, it is not clear to the reader without the species list. I suggest that the authors present a species list organized by order, then family, giving the different habitats, indicating species as present or absent, and by what trapping type. Such data sets are critically important for any understanding of biodiversity and are very interesting to readers. There is a great deal of excellent work here, but the authors  need to tell the reader what was caught in clear plain terms, and I suggest without statistical analyses. This is an excellent body of research of importance, but written in a way that makes it hard for readers to access. I again reiterate that this work is excellent, species lists are critically important and need to be published. The work done is impressive, and providing more complete information as outlined above would be a real benefit to our understanding of biodiversity. The overall merit is for the work done.

This needs to be clear for readers. For example the simple summary opening needs to be stated in plain language. This paper assessed insects in a park, a very good and worthy goal. The reader wants to know what the paper is about, so I suggest starting with your third sentence: "In the present work we considered areas located in a park of 4000 ha in northern Italy . . . "

Author Response

Dear Rewiever, thank you for your comment and appreciation of our research. 

In the attached PDF you will find our point-by-point response and in the article PDF you will find all the changes.

Best regards

Daniela

Reviewer 2 Report

See attached.

A thorough review of the language by an English speaker in terms of spelling, grammar, and syntax is a must.

Author Response

Dear Rewiever, thank you for your comment and appreciation of our research. 

In the attached PDF you will find our point-by-point response and in the article PDF you will find all the changes.

Best regards

Round 2

Reviewer 2 Report

Although there are improvements, the English does require more work (see attached).

Please see attached document. Nearly all highlighted areas (few exceptions) refer to spelling, grammar, and syntax in need of revision.

Author Response

Dear Reviewer, thank you for your precious comments, we changed all the parts signalled and made amendament to the text 

yours faithfully

the Authors